# Inorganic Elements in *Mytilus galloprovincialis* Shells: Geographic Traceability by Multivariate Analysis of ICP-MS Data

**DOI:** 10.3390/molecules26092634

**Published:** 2021-04-30

**Authors:** Tiziana Forleo, Alessandro Zappi, Dora Melucci, Martina Ciriaci, Francesco Griffoni, Simone Bacchiocchi, Melania Siracusa, Tamara Tavoloni, Arianna Piersanti

**Affiliations:** 1Department of Chemistry, University of Bari “Aldo Moro”, 70125 Bari, Italy; tiziana.forleo@uniba.it; 2Department of Chemistry “Giacomo Ciamician”, University of Bologna, 40126 Bologna, Italy; alessandro.zappi4@unibo.it; 3Istituto Zooprofilattico Sperimentale dell’Umbria e delle Marche “Togo Rosati”, 60131 Ancona, Italy; m.ciriaci@izsum.it (M.C.); f.griffoni@izsum.it (F.G.); s.bacchiocchi@izsum.it (S.B.); m.siracusa@izsum.it (M.S.); t.tavoloni@izsum.it (T.T.); a.piersanti@izsum.it (A.P.)

**Keywords:** *Mytilus galloprovincialis*, trace metals, ICP-MS, mussel, traceability, chemometrics, geographic origin

## Abstract

The international seafood trade is based on food safety, quality, sustainability, and traceability. Mussels are bio-accumulative sessile organisms that need regular control to guarantee their safe consumption. However, no well-established and validated methods exist to trace mussel origin, even if several attempts have been made over the years. Recently, an inorganic multi-elemental fingerprint coupled to multivariate statistics has increasingly been applied in food quality control. The mussel shell can be an excellent reservoir of foreign inorganic chemical species, allowing recording long-term environmental changes. The present work investigates the multi-elemental composition of mussel shells, including Al, Cu, Cr, Zn, Mn, Cd, Co, U, Ba, Ni, Pb, Mg, Sr, and Ca, determined by inductively-coupled plasma mass-spectrometry in *Mytilus galloprovincialis* collected along the Central Adriatic Coast (Marche Region, Italy) at 25 different sampling sites (18 farms and 7 natural banks) located in seven areas. The experimental data, coupled with chemometric approaches (principal components analysis and linear discriminant analysis), were used to create a statistical model able to discriminate samples as a function of their production site. The LDA model is suitable for achieving a correct assignment of >90% of individuals sampled to their respective harvesting locations and for being applied to counteract fraud.

## 1. Introduction

Fish products are a valuable source of nutrients for humans. Global consumption of seafood increased steadily from 1961 to 2017, at an annual rate of 3.1%, and aquaculture became a fundamental source of fish production, contributing up to 46% of the total fish market. Among marine animal organisms, bivalve mollusks represent an inexpensive source of protein, characterized by high biological value, and also providing essential minerals, trace metals, and vitamins [1]. Mussels, as a result of their filter-feeding habit and sessile lifestyle, tend to accumulate substances from the surrounding environment; hence, they may be considered as natural “devices” enabling pollution biomonitoring [2].

The international seafood trade requires food safety, quality, sustainability, and traceability standards. As a result of mussel’s bio-accumulative ability, regular control is necessary to guarantee their safe consumption [3] in compliance with Regulations (EC) 852/2004, 853/2004, and 854/2004 of the European Parliament [4,5,6,7].

Mussel bioaccumulation of metals, persistent organic contaminants, and biological contaminants is also a very useful marker of environmental pollution, reflecting, in some cases, the impact of anthropogenic activities on ecosystems [8,9,10,11].

Both the shells and tissues accumulate inorganic contaminants and may be used as indicators; however, several studies have stated that only the shells provide a historical memory of metal content throughout the organism lifetime, which is also preserved after the organism’s death [9,10]

The mussel shell is a complex organic/inorganic system consisting of two calcified valves coated by an outer organic layer, the periostracum, which supports the initial nucleation for the calcareous shell growth. The shell accumulation of trace metals starts from the cells of a mussel region, the mantle, which secretes the periostracum. A layer of calcium carbonate is laid down on it. As the mollusk grows, the epithelial cells in the inner cavity of the mantle gather calcium and bicarbonate ions from the water, enabling shell building. Other metals, like Mg, Mn, Zn, Pb, and Cu, can substitute the Ca^2+^ and be incorporated in the crystals. Almost any metal present in the environment can be taken up, actively metabolized by the organism, and incorporated into the carbonaceous structure [8,12].

The composition of mussel tissues depends not only on the geographical origin, but also on weight, dimensions, and spatial-environmental distribution [13,14,15].

Several studies have shown that shells and soft tissues have different levels of inorganic element bioaccumulation because, although the shell may incorporate much fewer metals than the soft tissues, their metabolization is lower, enabling element sequestration [16]. The shell chemical composition is strongly influenced by external variables, such as water temperature, salinity, pH, dissolved oxygen, sea currents, and coastal pollution, which can influence the growth rate, the calcium carbonate crystallization, and the possible inclusion of trace elements in the shell structure. The shell can be an excellent reservoir for the accumulation of foreign inorganic chemical species, allowing even long-term environmental changes to be recorded [2,17,18].

Given all the above statements, the present study aimed to exploit the mussel’s inorganic composition to trace their geographic origin. Therefore, mussel contaminant monitoring, paired with their inorganic fingerprints, may help in the safety tracing of mussels.

An analytical method enabling the analysis of a large set of inorganic elements in mussel shells was optimized by inductively-coupled plasma mass-spectrometry (ICP-MS), and the most relevant elements to be used for a possible geographical origin identification were selected. The elements were analyzed in several mussel shells, and the obtained results were processed by chemometric tools like principal component analysis (PCA) and linear discriminant analysis (LDA). The goal was to create a statistical model able to identify the sample’s geographic origin by means of the inorganic and trace element composition of their shells. To the best of our knowledge, this is the first work dealing with the geographic traceability of *Mytilus galloprovincialis* collected in the Adriatic Sea.

## 2. Results and Discussion

Many papers have analyzed trace elements in the soft tissue and byssus of mussels to define temporal and spatial variation [19,20,21]. Recently, some studies have also focused on shell inorganic composition, because it provides a stable, long-term, site-specific marker of elements, which is strongly influenced by the environment. This specific fingerprint may be useful for geographical-origin traceability [17,18,22,23].

### 2.1. Analytical Method

Mussels shell pre-treatment and acid digestion needed to be optimized to obtain mineralized solutions free from interferences.

Experiments were performed in triplicate to optimize the best pre-treatment time (2, 4, 6, and 10 days) by H_2_O_2_ (30%) soaking. After 2, 4, and 6 days, not all the organic material and periostracum seemed to be exhaustively removed from the shell surfaces. Only after 10 days were the valves perfectly clean and easily crunchable, yielding a fine homogenous powder.

Microwave sample digestion experiments were performed to obtain a clear mineralized solution. Initially, shell samples were digested using nitric and hydrofluoric acid (according to the EPA Method 3052, [24]), but a solid precipitate appeared in the solution (probably the low soluble CaF_2_ salt). Other acidic mineralization mixtures were tested without satisfactory results, especially regarding calibration linearity and repeatability. The best conditions were those obtained following EPA Method 3051A [25].

With the optimized conditions, the presented method showed good linearity (determination coefficients R^2^ > 0.990) and low response factor variability (accepted ±20%). Good trueness and precision were reached in reference material analysis (CRM recovery: 73–129%, only Al = 63%; RSD (%): 3–24%, only Al = 30%). 

### 2.2. Chemometrics

Input data are organized in datasets (matrices), whose rows correspond to sample replicates (objects), while the columns correspond to the measured variables. Each column corresponds to a metal concentration (overall data matrix is available as Appendix A).

A Pearson correlation matrix for the farmed shells, reported in Table 1, describes the associations between variables. Calcium was highly and directly correlated to Ni (0.79). There were other significant correlations, positive between Co and Cu (0.56) and negative between Mg and Ca (−0.50), and between Mg and Ba (−0.62). Such correlations may indicate the coexistence of Ca and Ni in mussels; perhaps these metals may be taken up using an analogous mechanism by the organism. Mg, instead, seems to compete with Ca as the structural material of the mussel shell.

The correlation matrix in Table 2 shows the correlations and anti-correlation between elements in the shells collected from natural banks.

The strongest anti-correlation was between Sr and Ni (−0.80). The correlations and anti-correlations between Zn and U (0.60), Ca and Co (−0.60), Mn and Al (0.57), Ca and Ni (0.56), Co and Mg (0.52), Cd and U (0.51), and Cd and Ba (0.50) were significant but less strong.

#### 2.2.1. Principal Component Analysis

A PCA model was computed for the matrix containing data from farmed mussel shells (153 rows × 14 columns). The first two PCs explain 36.9% of the total variance. Figure 1a shows the scores plot in the plane PC1 vs PC2. Figure 1b shows the loadings in the same space of the first two PCs. In the scores plot, letters identify the sampling sites and colors the sampling areas.

The influence of the different parameters can be evaluated from both the scores plot and the loadings plot. Except for three objects (three replicates of the same samples) in sampling sites “T” and “M”, both characterized by high levels of Sr and Ba, Figure 1a shows a good grouping of the sampling sites. Shells characterized by high levels of Ni and Ca are located to the right side of the graph; samples with high levels of Mg, Cu, Co, Mn, and Zn are located to the left side of the graph. However, it can also be observed that sampling sites close to each other are often placed on the opposite sides of the scores plot, as for “E” and “D”, both close to the city of Fano.

The factors affecting the grouping are unknown and could be related to the different water depths, or the different locations of the production plants with regards to the coast, including the relative position with respect to a river mouth, which may strongly influence the inorganic element water loading.

Another PCA model was created for the dataset containing mussels from natural banks (63 objects × 14 variables).

For the natural bank shells, Figure 2a shows the distribution of the objects and Figure 2b the distribution of the variables in the space PC1 vs. PC2. The first two PCs explain 43.9% of the total variance.

In this case, samples from the same production site are not always well grouped. This is particularly true for the samples coming from the Pesaro area (red samples in Figure 3a), where no clear grouping is highlighted. This could probably be the result of mussels growing on natural cliffs, which makes their shell’s inorganic composition more inhomogeneous, within the same sampling site, rather than what happens in controlled farming, which leads to greater homogeneity.

#### 2.2.2. Linear Discriminant Analysis

A classification method (LDA) was applied to the data in order to further highlight the eventual clusters that emerged from the PCA.

##### Farms and Natural Banks

The first analysis was applied to all samples (dataset with 216 objects × 14 variables), and considering the mussel origin (farmed or natural banks) as an a priori category. The LDA model showed an encouraging NER of 93.5%, with only three farmed objects misclassified as natural and 11 natural objects misclassified as farmed (5 from sampling point “V” and 6 from “Y”). The LDA model gave a 98.0% correct assignment for farmed, and 82.5% for natural growing mussels. Table 3 shows the confusion matrix, sensitivity, specificity, and NER for such an LDA model.

Next, data were divided into “farmed” and “natural” samples, in order to check, by LDA, if a stronger discrimination than that observed by PCA was present between sampling sites.

##### Farmed Samples

A LDA model was calculated for the farmed samples (dataset with 153 objects × 14 variables). The sampling sites were considered the a priori category. The classification method applied to this dataset gave an overall NER of 94.1%: only nine samples out of 153 were misclassified, assigning them to different sampling sites. As shown in the discriminant plot in Figure 3a, two samples collected in “B” sampling sites were misclassified as “C”, one sample from “F” as “E”, one sample from “H” as “R”, two samples from “M” as “G”, two samples from “P” as “Q”, and one sample from “R” as “I”.

Figure 3b shows the corresponding LDA-loadings plot, from which it can be seen that the most important variables for the discrimination were Co, Sr, Ni, Mn, and Zn, those furthest from the origin.

The values of sensitivity and specificity (Table 4), which were always higher than 66.7% (one misclassified sample may weigh for 10–20% if a class carries 9 or 6 samples), confirmed the good performance of the classification model.

##### Samples from Natural Banks

A LDA model was created from the dataset of natural bank mussel shells (dataset with 63 objects × 14 variables). Similarly, in this case, the a priori category was given to the sampling sites.

Figure 4 and Table 5 describe the quality of the classification model. The NER reached the value of 98.4%: only one sample out of 63 was wrongly assigned to another class (an object belonging to the “V” site was assigned to “Y”). The discriminant plot in Figure 4a confirms the good discrimination between all classes, with only a slight overlap between “K” and “U” classes (but without misclassification of samples between them). The LDA–loadings plot in Figure 4b shows that Co, Ni, Mg, and Sr are the most important variables for such a discrimination.

##### All Samples

A further LDA model was computed on the entire dataset, with both farmed and natural samples (dataset with 216 objects × 14 variables). The aim was to evaluate if the sampling sites could be discriminated, regardless of their production method (farmed or natural banks). Thus, in this case, the a priori categories were also the sampling sites. The LDA NER was again very good: 93.5%. Only 14 objects out of 216 were misclassified. Moreover, only three objects from the natural bank samples were misclassified into farmed shells, and only one object from the farmed shells was misclassified as a natural bank samples. The sensitivity and specificity (Table 6) for all 25 classes were higher than 62.5%. The discriminant plot in Figure 5a shows, besides the good discrimination of class “Z”, several overlaps between classes, thus, particularly in this case, it is important to evaluate the numerical results of LDA (Table 6) to assess the model performance. The LDA–loadings plot in Figure 5b shows that the most important variables for discrimination are Co, Ni, and Zn.

As described above, the implemented statistical model enabled a good classification performance, with a high percentage of correct assignment and efficient discrimination of farmed samples from wild ones. Mussels from natural banks, indeed, grow in an environment significantly different from farmed ones, thus also strongly influencing the aspect and composition of shells. First, the lifetime scale of natural mussels is usually longer and less standardized than farmed mussels, enabling the shells to grow thicker and more robust. Moreover, mussels from natural banks, being closer to the coast and having a larger nutrient availability, share their growing habitat with several other organisms, which also colonize their shells, unavoidably influencing their composition.

In Table 7 we compare our results with the ones obtained in other studies, based on ICP and dealing with the geographic traceability of bivalves by analyzing the mineral composition of their shells. In the table are reported: the species analyzed, the sampling sites and areas, the elements detected, the statistical methods applied, and the models correct assignment score (%). It emerged that the model developed in the present work had a good assignment performance (LDA: 98.4% for NB mussels, 94.1% for farmed mussels), and better than most of the other studies reported in Table 7. Comparable, and even slightly better, were the results obtained by M. Bennion et al. (2019) [18], even if the latter, to achieve a 100% correct assignment, needed to combine statistical modeling with the chemical composition of three structures (clean shell, foot, and periostracum) instead of analyzing only the clean shell, as we did.

## 3. Materials and Methods

### 3.1. Samples and Datasets

Mussels samples belonging to the *Mytilus galloprovincialis* species were harvested from 25 sites (18 farms and 7 natural banks) along the Marche region coast (central Italy), in the Adriatic Sea, in the framework of a regional surveillance plan. Natural banks are cliffs where mussels grow naturally, while farms are off-shore breeding sites, located roughly two nautical miles away from the coast. Figure 6 shows the geographical distribution of the sampling sites, and Table 8 summarizes the sampling periods and the number of samples for each site. European regulations request periodical monitoring of mollusk production areas, analyzing marine biotoxins, and microbiological and chemical contaminants [4,5,6]. The samples described here were selected from those of commercial size (>4 cm). Shell samples were taken between June and October 2018, and each site was generally sampled 3 times, for a total of 72 samples. Each one of the 72 samples was analyzed in triplicate, and 14 elements (Al, Cu, Cr, Zn, Mn, Cd, Co, U, Ba, Ni, Pb, Mg, Sr, Ca) were determined by ICP-MS, selecting those recognized from the bibliography as the most interesting to characterize geographical origin [22]. The results (72 samples × 3 replicates) were used to create a dataset whose columns were the concentration of the 14 elements (variables) and whose rows were the analysis replicates; therefore, the dimensions of the datasets were: 216 × 14 for the whole data, 153 × 14 for the farmed dataset (153 rows = 51 samples × 3 replicates, 14 variables), and 63 × 14 for the natural bank dataset (63 rows = 21 samples × 3 replicates, 14 variables).

### 3.2. Experimental

#### 3.2.1. Reagents, Standards, and Certified Reference Materials

The reagents used in sample digestion were Nitric acid (HNO_3_, 65%, Suprapur^®^), Hydrogen Peroxide (H_2_O_2_, 30% *_v/v_*, Super-pure, p.a., reag. ISO, reag. Ph. Eur.), Hydrofluoric acid (HF, 40%, AnalaR NORMAPUR^®^), and Hydrochloric acid (HCl, 37%, Super-pure^®^ RS).

Rhodium (Rh) and Lutetium (Lu) pure standard at 1000 μg mL^−1^ were used to prepare the Internal Standard Solution (Rh: 0.2 mg L^−1^ and Lu: 0.1 mg L^−1^ in 1%*_v_*_/*v*_ HNO_3_).

Matrix match calibration curves were prepared using Multi-Element Calibration Standard 3 (Al, As, Ba, Be, Bi, Cd, Ca, Cs, Cr, Co, Cu, Ga, In, Fe, Pb, Li, Mg, Mn, Ni, K, Rb, Se, Ag, Na, Sr, Tl, U, V, Zn: 10 μg mL^−1^; Perkin Elmer). Quality control spiked samples were prepared using Multi-Element Calibration Standard 3 (10 μg mL^−1^; Perkin Elmer), a solution of Ca, Mg (5000 mg L^−1^; CPAchemLDT), and mono-element stock solutions for Al (1000 mg L^−1^; CPAchemLDT), Ni, Mn, Zn (1000 µg mL^−1^; Perkin Elmer). All solutions were prepared in 5%*_v_*_/*v*_ HNO_3_.

The rinse solution was 5%*_v_*_/*v*_ HNO_3_. Milli-Q water grade was produced by a Millipore purification system (Bedford, MA, USA).

Certified reference material was included in the analytical batches in order to assess the precision and trueness (accuracy) of the method: Dolt 5, fish liver (NRC: National Research Council Canada). 

#### 3.2.2. Sample Pretreatment and Preparation

The mussel samples were mechanically opened, the soft tissues removed, and the shells rinsed with water. To analyze a sufficiently homogeneous and representative quantity of sample, 4 valves were submitted to pre-treatment by immersion in H_2_O_2_ 30% solution for 10 days [22]. The dried shell was then homogenized to obtain a powder. Microwave sample digestion was accomplished following the Environmental Protection Agency procedure 3051A: 0.25 g of homogenized shells was treated with 9 mL HNO_3_ 67–69% and 1 mL of HCl 37% in a high-pressure microwave system (Milestone-Ethos1-HPR1000), using a temperature/pressure programming procedure [25].

The thus obtained digested solution was diluted with ultrapure water, and adjusted to a final volume corresponding to 50 ± 1.0 g in polypropylene vessels. Further dilutions were performed according to the elements to be analyzed. 

#### 3.2.3. Instrumental Analysis

The analyses of Al, Ba, Cd, Ca, Co, Cr, Cu, Mg, Mn, Ni, Pb, Zn, Sr, and U were performed by ICP-MS, on an ELAN DRC II (Perkin Elmer) equipped with an ASX-520 autosampler (Perkin Elmer). The sample was introduced by a peristaltic pump, a Meinhard quartz concentric nebulizer, and a cyclonic quartz spray chamber. The internal standard solution was automatically delivered through a mixing block. Argon (purity 99.999%) was used as the plasma torch, nebulizer, and auxiliary gas, while ammonia (purity 99.99%) was the reaction gas in DRC mode. The ICP-MS parameters were (Table 9): 

The calibration curves used for quantification were matrix-matched from 0.1 µg L^−1^ to 500 µg L^−1^, choosing for each element the most appropriate range.

Each calibration curve was created on at least 3 points other than the matrix sample. The best fit curve was obtained by interpolating the response factors (counts sec^−1^ element/counts sec^−1^ internal standard) and the respective concentrations, and using the least-squares algorithm. 

The method performances were assessed by evaluating linearity, precision (repeatability), trueness, and quantification limits (LOQ). The last figure of merit was calculated based on signal-to-noise ratio, using the criterion 6σ [26]. The experimental values for LOQ were: 0.002 mg/kg for Cd, Pb, U; 0.020 mg/kg for Ba, Co, Cr; 0.040 mg/kg for Cu; 0.20 mg/kg for Mn, Ni, Sr; 0.50 mg/kg for Al, and Zn; 20 mg/kg for Ca, Mg. 

#### 3.2.4. Quality Assurance/Quality Control

The laboratory background contamination level was monitored by analyzing reagent blank samples; accuracy was assessed by batch certified reference material (CRM Dolt5) and spiked sample analysis.

The shell matrix was shown to be highly inhomogeneous, therefore batch-to-batch repeatability checking was necessary, and all the samples were analyzed in triplicate in order to have robust results.

### 3.3. Chemometrics

Pearson correlation matrix was calculated to evaluate correlation or anti-correlation between variables (metals concentration). A direct correlation takes place when the correlation index is positive, otherwise, the variables are anti-correlated [27]. When the correlation index between two variables is less than |0.3|, the two variables are considered weakly correlated. The correlation is considered strong when the correlation index is >|0.7|. When the index is between |0.3| and |0.7| the correlation is considered significant. This choice was empirically made by the authors, according to an already published method [28].

Principal components analysis is a well-known chemometric technique used for the exploratory analysis of a data structure [29]. It consists of projecting the data into a subspace whose axes are called principal components (PCs). The PCs are obtained from a linear combination of the original variables and contain a decreasing amount of information, when moving from the first to the last PC [30]. The most important PCA outputs are the scores and the loadings plots, which show respectively the objects (rows of the dataset) and the variables (columns of the dataset) in the PC space, and allow studying the relationships between them. These graphical models are used to verify whether the objects form any clusters, which would be an indication of possible discrimination power in the variables, and whether the clusters agree with the eventual assignment of samples to a priori categories. When PCA shows clustering, it makes sense to go on attempting the creation of multivariate classification models, in a mathematical and not only graphical form. 

Linear discriminant analysis (LDA) is a classification approach based on the maximization of the ratio of the between-class variance to the within-class variance in the dataset, thereby pursuing the maximal discrimination between known sample classes [31]. It assumes that all variables have a Gaussian distribution and are statistically independent. A first validation of the model is carried out by cross-validation (CV) [32]. While PCA is an unsupervised method that allows defining a representation of the data in an orthogonal space without any hypothesis about the belonging of the objects to a given class, LDA is a supervised method that, starting from a priori known classes, aims to create a classification model based on a certain number of independent variables.

Several parameters can be used for the quality estimation of the classification models [33]. All indexes can be derived from the confusion matrix, which is a square matrix with dimensions *G* × *G*, where *G* is the number of a priori classes. The diagonal elements *n_gg_* represent the correctly classified objects, while the off-diagonal elements represent the objects erroneously classified.

The non-error rate (NER) is a measure of the quality of a classification model: it represents the percentage of correctly assigned objects. The NER is defined as follows (Equation (1)):(1)$$NER=\frac{\sum_{g=1}^{G}n_{gg}}{n}$$
where *n* is the total number of objects. It is also called accuracy, or classification rate.

The sensitivity (*Sn_g_*) describes the model’s ability to recognize objects belonging to the *g*-th class and is defined as (Equation (2)):(2)$$Sn_{g}=\frac{n_{gg}}{n_{g}}$$
where *n_g_* is the number of objects belonging a priori to the *g*-th class. 

The specificity (*Sp_g_*) is the ability of the *g*-th class to reject the objects of all other classes and is defined as (Equation (3)):(3)$$Sp_{g}=\frac{n_{gg}}{{n^{\prime }}_{g}}$$
in which *n_g_’* is the total number of objects assigned to the *g*-th class [27]. 

NER, sensitivity, and specificity can assume values between 0 (no class discrimination) and 1 (perfect class discrimination) [34].

Person’s correlation matrices and the PCA were calculated by the software CAT [35], based on the R environment (R Core Team, Vienna, Austria). LDA was performed using the R software.

## 4. Conclusions

A method enabling the simultaneous ICP-MS analysis of 14 elements (Al, Cu, Cr, Zn, Mn, Cd, Co, U, Ba, Ni, Pb, Mg, Sr, Ca) in mussel shells after microwave digestion was developed and optimized. The method was applied to mussel shells collected in 25 sampling sites located along the central Adriatic Sea coast, and the obtained results were processed by chemometric tools, in order to discriminate mussels as a function of their production method and site.

Among the here applied statistical methods, LDA models were able to achieve a high percentage (>90%) of correct assignments of the samples to a specific sampling point. Therefore, this model could be applied as a useful tool to ensure mussel traceability, counteract fraud, and guarantee product quality.

Moreover, this work indicates that the contents of Al, Cu, Cr, Zn, Mn, Cd, Co, U, Ba, Ni, Pb, Mg, Sr, and Ca in the shells of *Mytilus galloprovincialis* have discriminating power concerning the production method and sampling site. In particular, LDA showed a potentially interesting discriminating power for Co, Ni, Zn, and Sr, due to the high LDA loadings shown in all the computed models. However, to obtain a robust multivariate screening method, further work is needed: the dataset will be implemented with a higher number of samples for each site and for each sampling period, whose eventual influence on results has not yet been explored.

## Figures and Tables

**Figure 1 molecules-26-02634-f001:**
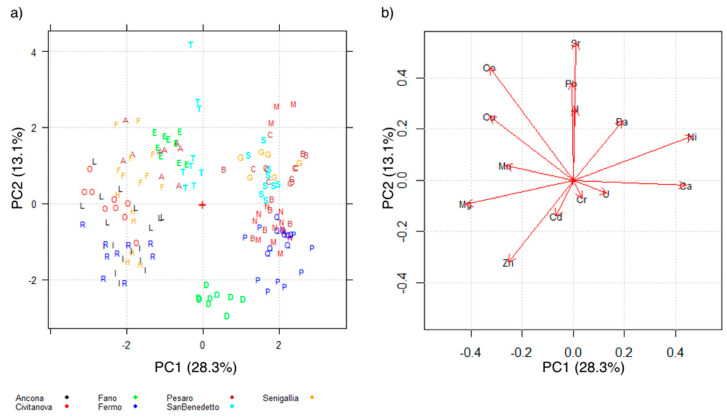
(**a**) Scores plot of farmed shell samples. Letters indicate the sampling sites and colors the sampling areas; (**b**) Loadings plot of the variables in the space of the first two PCs.

**Figure 2 molecules-26-02634-f002:**
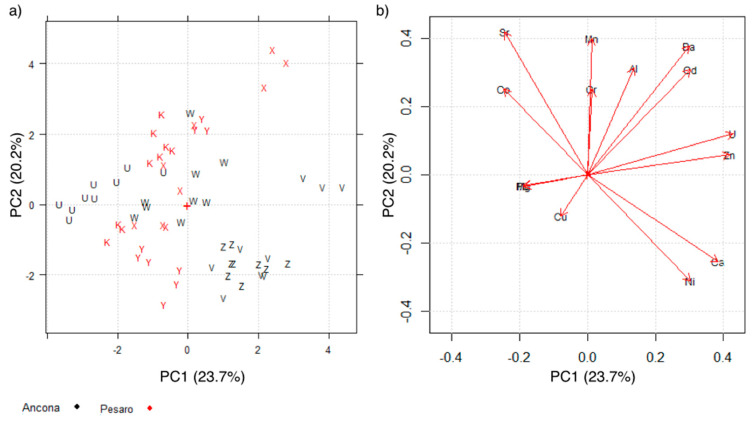
(**a**) Scores plot of shell samples collected from natural bank shell samples. Letters indicates the sampling sites, colors the sampling areas; (**b**) Loadings plot of the variables in the space of the first two PCs.

**Figure 3 molecules-26-02634-f003:**
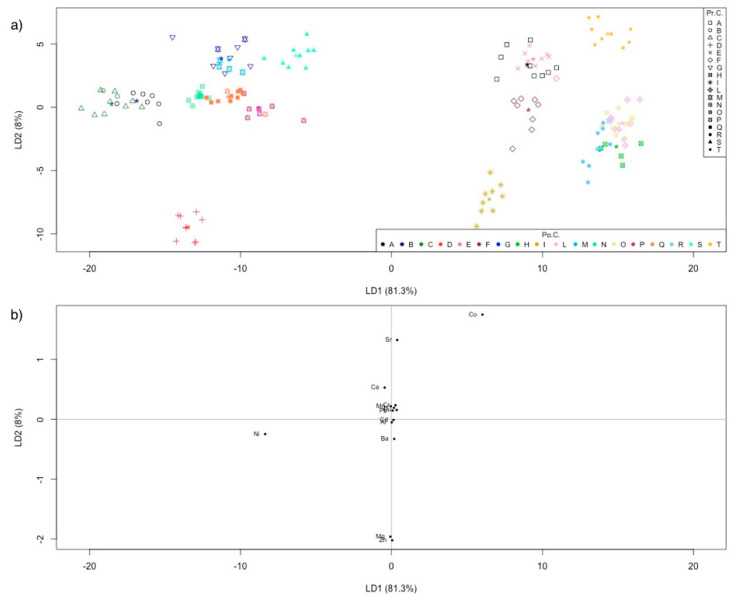
(**a**) Farmed shells discrimination plot. LDA is calculated using the sampling site as a priori category. Symbols and colors represent the prior classes and the posterior classes, respectively. The asterisk is the centroid of the class. “Pr.C.” are the prior classes, “Po.C.” are the posterior classes; (**b**) LDA–loading plot for the farmed shells.

**Figure 4 molecules-26-02634-f004:**
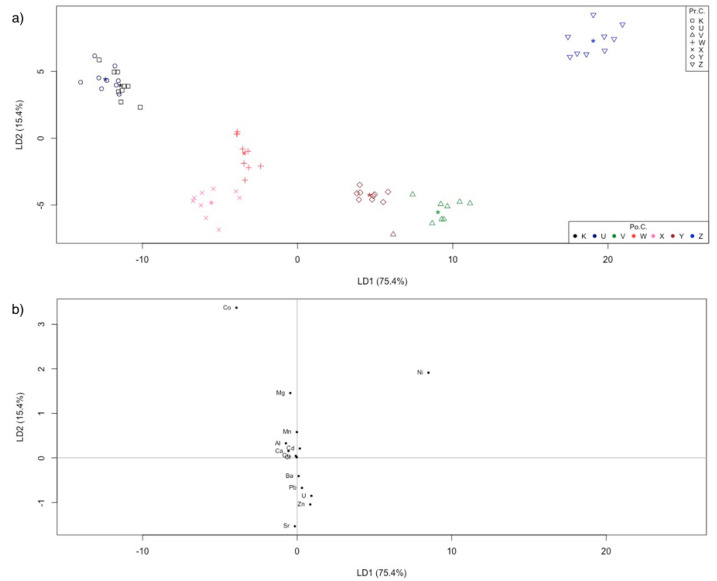
(**a**) Natural bank samples discrimination plot. LDA was calculated using the sampling sites as the a priori categories. Symbols and colors represent the prior classes and the posterior classes, respectively. The asterisk is the centroid of the class. “Pr.C.” are the prior classes, “Po.C.” are the posterior classes; (**b**) LDA–loadings plot for natural bank samples.

**Figure 5 molecules-26-02634-f005:**
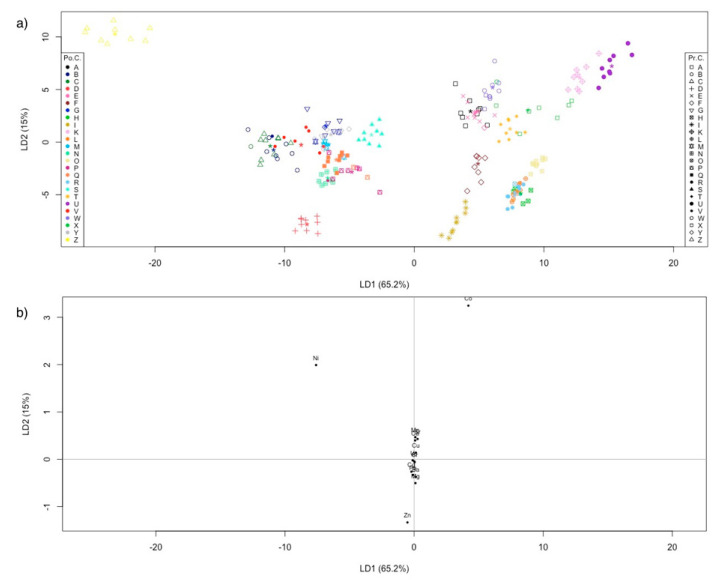
(**a**) Farmed and natural bank samples discrimination plot. LDA was calculated using the sampling sites as a priori categories. Symbols and colors represent the prior classes and the posterior classes, respectively. The asterisk is the centroid of the class. “Pr.C.” are the prior classes, “Po.C.” are the posterior classes; (**b**) LDA–loading plot for farmed and natural bank samples.

**Figure 6 molecules-26-02634-f006:**
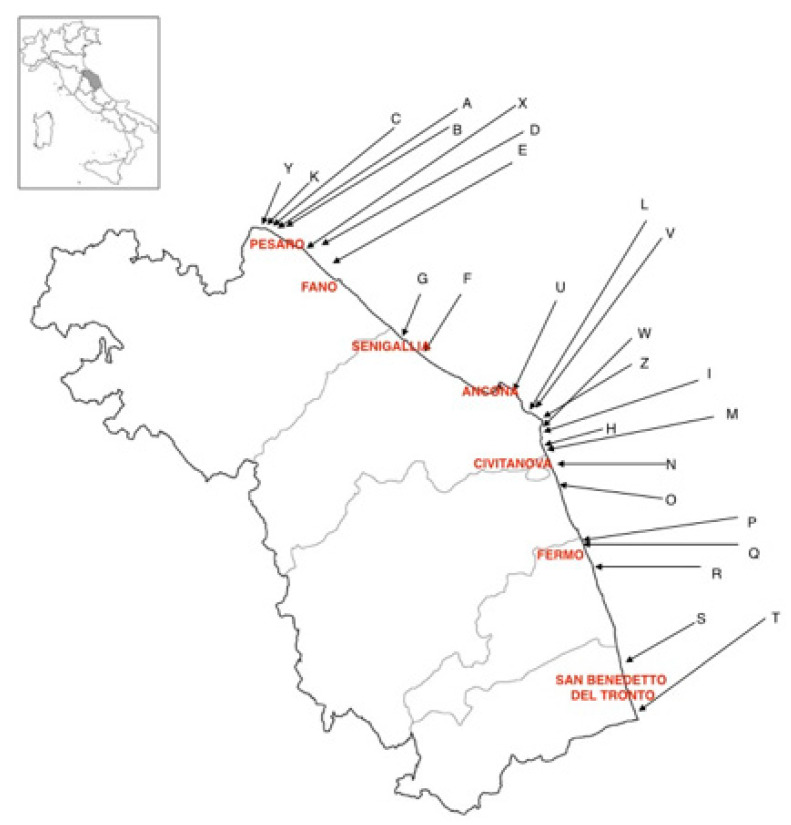
Farm and natural bank geographical distribution.

**Table 1 molecules-26-02634-t001:** Pearson correlation matrix for metals in farmed mussel shells.

	Al	Cu	Cr	Zn	Mn	Cd	Co	U	Ba	Ni	Pb	Mg	Sr	Ca
**Al**	1.00	0.19	−0.06	−0.01	−0.04	−0.08	0.11	−0.13	−0.01	0.15	0.05	−0.08	0.13	0.04
**Cu**		1.00	0.08	0.18	0.31	−0.01	**0.56**	0.01	−0.12	−0.31	0.09	0.36	0.05	−0.32
**Cr**			1.00	−0.05	−0.03	0.07	0.01	−0.07	0.08	0.04	−0.18	−0.1	−0.07	0.05
**Zn**				1.00	0.41	−0.17	−0.12	0.06	−0.21	−0.36	−0.11	0.35	−0.24	−0.28
**Mn**					1.00	−0.00	0.14	−0.00	−0.08	−0.18	0.41	0.45	−0.03	−0.17
**Cd**						1.00	0.04	0.01	0.14	−0.25	0.00	0.09	−0.26	−0.14
**Co**							1.00	−0.17	−0.08	−0.40	0.18	0.30	0.34	−0.52
**U**								1.00	0.37	0.23	−0.14	−0.03	0.02	0.21
**Ba**									1.00	0.31	0.05	−0.18	0.30	0.10
**Ni**										1.00	0.17	−**0.62**	0.1	**0.79**
**Pb**											1.00	−0.14	0.14	0.08
**Mg**												1.00	0.01	−**0.50**
**Sr**													1.00	−0.08
**Ca**														1.00

**Table 2 molecules-26-02634-t002:** Pearson correlation matrix for metals in shells belonging to mussels from natural banks.

	Al	Cu	Cr	Zn	Mn	Cd	Co	U	Ba	Ni	Pb	Mg	Sr	Ca
**Al**	1.00	0.05	−0.01	0.15	**0.57**	0.29	−0.07	0.29	0.44	−0.10	0.18	0.10	0.15	0.03
**Cu**		1.00	−0.06	−0.00	−0.18	−0.09	0.06	−0.03	−0.27	−0.01	0.08	0.14	−0.14	−0.12
**Cr**			1.00	−0.00	0.17	0.16	0.21	0.20	0.18	−0.18	−0.22	−0.09	0.20	−0.16
**Zn**				1.00	−0.12	0.42	−0.17	**0.60**	0.48	0.16	−0.29	−0.28	−0.22	0.44
**Mn**					1.00	0.21	0.16	0.11	0.50	−0.07	0.28	0.07	0.31	−0.20
**Cd**						1.00	−0.05	**0.51**	**0.50**	−0.01	−0.26	−0.33	0.19	0.09
**Co**							1.00	−0.06	0.11	−0.30	−0.05	**0.52**	0.31	−**0.60**
**U**								1.00	0.45	0.33	−0.33	−0.09	−0.33	0.35
**Ba**									1.00	0.10	−0.13	−0.09	0.17	0.15
**Ni**										1.00	−0.08	0.11	−**0.80**	**0.56**
**Pb**											1.00	0.19	−0.02	−0.10
**Mg**												1.00	−0.10	−0.11
**Sr**													1.00	−0.49
**Ca**														1.00

**Table 3 molecules-26-02634-t003:** Confusion matrix, sensitivity (Sn), and specificity (Sp), and NER values for the a priori classes: shells from farms (F) and natural banks (NB).

	Predicted (CV)			
Actual	F	NB	Sn	Sp
**F**	150	3	0.980	0.932
**NB**	11	52	0.825	0.945
			**NER**	0.935

**Table 4 molecules-26-02634-t004:** Farmed site LDA, sensitivities (Sn), and specificities (Sp) of the model.

**Sampling Site**	**A**	**B**	**C**	**D**	**E**	**F**	**G**	**H**	**I**
**Sn**	1.00	0.778	1.00	1.00	1.00	0.889	1.00	0.833	1.00
**Sp**	1.00	1.00	0.818	1.00	0.900	1.00	0.750	0.833	1.00
**Sampling Site**	**L**	**M**	**N**	**O**	**P**	**Q**	**R**	**S**	**T**
**Sn**	1.00	0.667	1.00	1.00	0.778	1.00	0.889	1.00	1.00
**Sp**	1.00	1.00	1.00	1.00	1.00	0.818	0.889	1.00	1.00

**Table 5 molecules-26-02634-t005:** Natural bank sites LDA, sensitivities (Sn), and specificities (Sp) of the model.

Sampling Site	K	U	V	W	X	Y	Z
**Sn**	1.00	1.00	0.889	1.00	1.00	1.00	1.00
**Sp**	1.00	1.00	1.00	1.00	1.00	0.900	1.00

**Table 6 molecules-26-02634-t006:** Overall sampling site LDA, sensitivities (Sn), and specificities (Sp) of the model.

**Sampling Site**	**A**	**B**	**C**	**D**	**E**	**F**	**G**	**H**	**I**
**Sn**	1.00	0.889	1.00	1.00	1.00	0.889	0.833	0.833	1.00
**Sp**	1.00	0.889	0.900	1.00	0.900	1.00	0.625	1.00	1.00
**Sampling Site**	**K**	**L**	**M**	**N**	**O**	**P**	**Q**	**R**	
**Sn**	1.00	1.00	0.667	1.00	1.00	0.778	1.00	1.00	
**Sp**	1.00	1.00	1.00	1.00	1.00	0.875	0.818	0.900	
**Sampling Site**	**S**	**T**	**U**	**V**	**W**	**X**	**Y**	**Z**	
**Sn**	1.00	1.00	1.00	0.778	0.889	0.778	0.889	1.00	
**Sp**	1.00	1.00	1.00	1.00	0.800	0.875	0.889	1.00	

**Table 7 molecules-26-02634-t007:** Literature review: bivalves trace element fingerprint (species, geographical areas, elements, statistical models, assignment rate).

Matrix	Species	Areas (Site) (Analysed Elements)	Statistical Model *	Correct Assignment Rate (%)	Reference
shells, foot, periostracum	*Mytilus edulis*	Ireland coast (4) (As, Cd, Cu, Fe, Mn, Ni, Pb, Sr, Zn )	Random forest analysis	67.5 – periostracum 100 − shells + foot + *periostracum*	M. Bennion et al., 2019 [18]
shells	*Amblema plicata,* *Quadrula quadrula*	North American river (5) (Mn, Fe, Co, Ni, Cu, Zn, As, Cd, Se)	ANOVA PCA		W. A. Wilson et al., 2018 [2]
shells	*Cerastoderma edule*	Portuguese Atlantic Coastline (8) (Mg/Ca, Mn/Ca, Sr/Ca, Ba/Ca)	MANOVA LDA	90	F. Ricardo et al., 2017 [17]
shells	*Cerastoderma edule*	Estuarine system Ria de Aveiro, Portugal (5) (Ba/Ca, Mg/Ca, Mn/Ca, Pb/Ca, Sr/Ca)	ANOSIM ANOVA SIMPER CAP	92	F. Ricardo et al., 2015 [22]
shells (juveniles and adults)	*Mytilus edulis*	Gulf of Maine, USA (7) (Ba/Ca, Cu/Ca, Pb/Ca, La/Ca, Sr/Ca, Mg/Ca, Zn/Ca)	LDA ANOVA	68.4−juvenile mussels 57.3−adult mussels	C. J. B. Sorte et al., 2013 [23]
shells	*Mytilus gallop.*	Central Adriatic Sea Coast (Marche Region) (25: 18 farmed, 7 natural banks) (Al, Cu, Cr, Zn, Mn, Cd, Co, U, Ba, Ni, Pb, Mg, Sr, Ca)	PCA LDA	98.4−natural banks mussels 94.1−farmed mussels	This paper

***** ANOVA = Analysis of variance, MANOVA = Multivariate analysis of variance, ANOSIM = Analysis of similarity, SIMPER = Similarity percentages, CAP = Canonical analysis of principal coordinates.

**Table 8 molecules-26-02634-t008:** Sampling areas, sites, and period.

	Sampling Area	Sampling Site	Sampling Period			Total Number of Samples
			1	2	3	
**Farmed**	Pesaro	A	July	July	July	3
		B	June	July	October	3
		C	July	October	October	3
	Fano	D	July	July	August	3
		E	July	August	September	3
	Senigallia	F	July	August	October	3
		G	July	August		2
		H	August	October		2
	Ancona	I	July	July	August	3
		L	July	July	September	3
	Civitanova	M	July	July		2
		O	July	July	August	3
		N	July	July	July	3
	Fermo	P	July	July	August	3
		R	July	July	August	3
		Q	July	July	August	3
	San Benedetto del Tronto	S	July	August	August	3
		T	July	August	August	3
**Natural Banks**	Ancona	U	July	July	August	3
		V	July	July	August	3
		Z	July	July	August	3
		W	July	July	August	3
	Pesaro	K	July	July	August	3
		Y	July	July	August	3
		X	July	July	August	3
					**Total**	72

**Table 9 molecules-26-02634-t009:** Optimized ICP-MS acquisition method.

ICP-MS PARAMETERS					
**Plasma gas (L/min)**		15	**Sweeps/reading**		20
**AUX gas (L/min)**		0.8 ÷ 1.5	**Readings/replicate**		1
**RF power (W)**		1300	**Replicates**		3
**Nebulizer gas (L/min)**		0.75 ÷ 1.04	**Scan mode**		Peak hopping
**Rpa**		0	**Dwell time (ms)**		100
**Sample uptake (mL/min)**		0.96	**Dwell time Pb, Cd, U (ms)**		200
	***m***/***z***	**Internal Standard**	**Acquisition Mode**	**RPq**	**Cell Gas: NH_3_ (mL/min)**
**Ba**	138	^103^Rh	Std mode	0.25	0
**Cd**	111				
**Co**	59				
**Ni**	60				
**Pb**	206 + 207 + 208	^175^Lu			
**Sr**	88	^103^Rh			
**U**	238				
**Al**	27	^103^Rh	DRC	0.75	0.5
**Mg**	24			0.45	0.7
**Ca**	44			0.5	0.5
**Cr**	52			0.35	0.5
**Mn**	55			0.35	0.5
**Cu**	63			0.75	0.45
**Zn**	68			0.75	0.45

## Data Availability

The data presented in this study are available as Appendix A.

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
