# Peer review of "Inorganic Elements in Mytilus galloprovincialis Shells: Geographic Traceability by Multivariate Analysis of ICP-MS Data"

_molecules, 2021, doi:10.3390/molecules26092634_

Round 1

Reviewer 1 Report

The manuscript by Forleo et al. focused on the determination of inorganic elements in mussel shells. The manuscript is well prepared. However, I have a few comments. Addressing them should improve the review:

  1. In my opinion, the discussion should be improved. The description of the obtained results dominates in the present manuscript.
  2. I want the Authors to explain how was calculated the number of components in the PCA analysis. Strangely, the first component has 13.1% and the second one 23.8%.
  3. Table 7 is not available in whole.
  4. Lines 341-344: on what basis the Authors determined the correlation index interval.
  5. Did three replications allow to determine the repeatability of the tested substances in the material?
  6. Why did the authors not show parameters such as LOD and LOQ when validating the method?

Author Response

The manuscript by Forleo et al. focused on the determination of inorganic elements in mussel shells. The manuscript is well prepared. However, I have a few comments. Addressing them should improve the review.

  • In my opinion, the discussion should be improved. The description of the obtained results dominates in the present manuscript.

AUTHORS: In fact, we focused mostly on the discussion of the results because, in our opinion, the dataset and the “explored” region are not large enough to draw general conclusions about the mussels or metals behavior, as we stated in the “conclusions” section: “However, to obtain a robust multivariate screening method, further work is needed: the dataset will be implemented with a higher number of samples for each site and for each sampling period, whose eventual influence on results could not be explored yet.”. Therefore, our aim is to expand the dataset, both as number of samples and as sampling points and to write a further, more general, work in the future.

  • I want the Authors to explain how was calculated the number of components in the PCA analysis. Strangely, the first component has 13.1% and the second one 23.8%.

AUTHORS: We thank the referee for its comment. In Figure 1 there was a typo error on the axis text: the explained variance of the PC1 was exchanged with the one of the PC2. We corrected Figure 1.

  • Table 7 is not available in whole.

AUTHORS: We modified Table 7 to make it more readable

  • Lines 341-344: on what basis the Authors determined the correlation index interval.

AUTHORS: We thank the referee for its comment. The choice of reference intervals for correlation coefficients is commonly used in analytical chemistry as a rule of thumb, and it has already been subjectively adopted by the authors. We have added this specification in the text (see lines 346-347 in the revised version).

  • Did three replications allow to determine the repeatability of the tested substances in the material?

AUTHORS: Three replicates enable the relative standard deviation calculation and the experimentally obtained results, for all the samples analyzed, were compliant with Horwitz-Thomson theoretical reference values, which means that the analysis had a good repeatability.

  • Why did the authors not show parameters such as LOD and LOQ when validating the method?

AUTHORS: We thank the reviewer for the comment, actually we missed out the LOQs which have been now introduced in paragraph “3.2.3. Instrumental analysis”.

Reviewer 2 Report

The manuscript studies specific inorganic elements composition of mussel shells collected along the Central Adriatic Coast. Their determination was performed by Inductively-Coupled Plasma Mass-Spectrometry. By means of  chemometrics (Principal Components Analysis and Linear Discriminant Analysis), authors proved that it was feasible to discriminate samples according to their production area. The research work is very interesting, well organized and it has scientific soundness. Therefore, my recommendation is to accept the manuscript after minor revision, after changing Table 7 concerning Literature review (it is not well presented).

Author Response

The manuscript studies specific inorganic elements composition of mussel shells collected along the Central Adriatic Coast. Their determination was performed by Inductively-Coupled Plasma Mass-Spectrometry. By means of chemometrics (Principal Components Analysis and Linear Discriminant Analysis), authors proved that it was feasible to discriminate samples according to their production area. The research work is very interesting, well organized and it has scientific soundness. Therefore, my recommendation is to accept the manuscript after minor revision, after changing Table 7 concerning Literature review (it is not well presented).

AUTHORS: We modified Table 7 to make it more readable

Round 2

Reviewer 1 Report

Well, the article seems much more attractive and interesting.